# *Graptopetalum paraguayense* Inhibits Liver Fibrosis by Blocking TGF-β Signaling In Vivo and In Vitro

**DOI:** 10.3390/ijms20102592

**Published:** 2019-05-27

**Authors:** Wei-Hsiang Hsu, Se-Chun Liao, Yau-Jan Chyan, Kai-Wen Huang, Shih-Lan Hsu, Yi-Chen Chen, Ma-Li Siu, Chia-Chuan Chang, Yuh-Shan Chung, Chi-Ying F. Huang

**Affiliations:** 1Institute of Biopharmaceutical Sciences, National Yang-Ming University, Taipei 11221, Taiwan; rabbitjim5@hotmail.com (W.-H.H.); eva900706@yahoo.com.tw (Y.-C.C.); 2Institute of Clinical Medicine, National Yang-Ming University, Taipei 11221, Taiwan; susanliao@everestpharm.com; 3Development of Natural Resource, Institute of Pharmaceutics, Development Center for Biotechnology (DCB), Taipei 11571, Taiwan; yjchyan5@dcb.org.tw (Y.-J.C.); malisiu@dcb.org.tw (M.-L.S.); yschung@dcb.org.tw (Y.-S.C.); 4Department of Surgery & Hepatitis Research Center, National Taiwan University Hospital, National Taiwan University, Taipei 10051, Taiwan; skyntuh@gmail.com; 5Department of Education and Research, Taichung Veterans General Hospital, Taichung 40705, Taiwan; l978830@hotmail.com; 6Department of Pharmacy, School of Pharmacy, National Taiwan University, Taipei 10050, Taiwan; chiachang@ntu.edu.tw

**Keywords:** *Graptopetalum paraguayense*, liver fibrosis, hepatic stellate cell, TGF-β

## Abstract

Background and Aims: Liver fibrosis is the excessive accumulation of extracellular matrix proteins, including collagen, which occurs in most types of chronic liver diseases. Advanced liver fibrosis results in cirrhosis, liver failure, and portal hypertension. Activated hepatic perivascular stellate cells, portal fibroblasts, and myofibroblasts of bone marrow origin have been identified as major collagen-producing cells in the injured liver. These cells are activated by fibrogenic cytokines, such as TGF-β1. The inhibition of TGF-β1 function or synthesis is a major target for the development of antifibrotic therapies. Our previous study showed that the water and ethanol extracts of *Graptopetalum paraguayense* (GP), a Chinese herbal medicine, can prevent dimethylnitrosamine (DMN)-induced hepatic inflammation and fibrosis in rats. Methods: We used rat hepatic stellate HSC-T6 cells and a diethylnitrosamine (DEN)-induced rat liver injury model to test the potential mechanism of GP extracts and its fraction, HH-F3. Results: We demonstrated that GP extracts and HH-F3 downregulated the expression levels of extracellular matrix (ECM) proteins and inhibited the proliferation and migration via suppression of the TGF-β1 pathway in rat hepatic stellate HSC-T6 cells. Moreover, the HH-F3 fraction decreased hepatic collagen content and reduced plasma AST, ALT, and γ-GT activities in a DEN-induced rat liver injury model, suggesting that GP/HH-F3 has hepatoprotective effects against DEN-induced liver fibrosis. Conclusion: These findings indicate that GP/HH-F3 may be a potential therapeutic agent for the treatment of liver fibrosis. The inhibition of TGF-β-mediated fibrogenesis may be a central mechanism by which GP/HH-F3 protects the liver from injury.

## 1. Introduction

Liver fibrosis results from chronic damage to the liver in conjunction with the accumulation of an extracellular matrix (ECM), which includes four large families of macromolecules: glycoproteins, collagens, proteoglycans, and carbohydrate [1,2]. Liver fibrosis is a dynamic process that usually occurs secondary to hepatic injury and inflammation, and progresses at different rates depending on the etiology of liver disease, and it is also influenced by environmental and genetic factors. Many etiologies produce liver fibrosis, including chronic hepatitis B virus (HBV) and hepatitis C virus (HCV) infection (HBV and HCV hepatitis, respectively), alcohol abuse (alcoholic steatohepatitis, ASH), nonalcoholic fatty liver disease (NAFLD), and nonalcoholic steatohepatitis (NASH). If fibrosis continues unopposed, it would disrupt the normal architecture of the liver, which alters the normal function of the organ, ultimately leading to excessive scarring and organ failure and resulting in liver cirrhosis [3].

A key step in the pathogenesis of liver fibrosis is the activation of hepatic stellate cells (HSCs) in response to chronic liver injury. This activation process is characterized by a phenotypic switch from a quiescent vitamin A-rich phenotype to a myofibroblastic fibrogenic phenotype; this switch is mainly due to the effects of several mediators, such as reactive oxygen species (ROS), lipid peroxidation (LPO) products, and fibrogenic cytokines such as transforming growth factor beta (TGF-β) and platelet-derived growth factor (PDGF). Several experimental animal models of liver disease also support the role of the HSC, which are perisinusoidal cells that normally reside in the space of disease in the liver and have stored lipid droplets, as the primary cell responsible for the excess of collagen production during fibrogenesis [4,5]. Therefore, HSCs have consistently been shown to play a key role in hepatic fibrogenesis, regardless of the underlying etiology [6].

Among the mediators involved in fibrogenesis, TGF-β is thought to play a central role and is the most potent fibrogenic cytokine described for HSCs; activated TGF-β stimulates the synthesis of ECM components, such as type I, III, and IV collagens, elastin, tenascin, osteonectin, biglycan, and decorin, and can also inhibit their degradation by inactivating matrix metalloproteinases (MMPs) through the upregulation of tissue inhibitors of metalloproteinases (TIMPs) [7,8]. The TGF-β1-activated Smad signaling and mitogen-activated protein kinase (MAPK) pathways stimulate experimental hepatic fibrosis and are potential targets for therapy [9,10]. Disrupting TGF-β synthesis and/or signaling pathways prevents scar formation in experimental liver fibrosis.

In Taiwan, *Graptopetalum paraguayense* (GP) is a medicinal plant that is regarded as a vegetable with health benefits. According to an archaic Chinese prescription, GP has been employed traditionally to remedy a number of pathological conditions including modulating blood pressure, alleviating hepatic disorders, relieving pains and infections, detumescence, and detoxication. The reported biological effects of GP include the inhibition of tyrosinase [11], antioxidants [12], hypertension regulation [13,14], and hepatoprotective effect [15]. We previously showed that GP extracts could inhibit liver cancer proliferation and have synergistic effects with sorafenib on hepatocellular carcinoma (HCC) cells [16,17]. Moreover, GP extract could not only suppress gluconeogenesis and lipogenesis, but also inhibit HBV gene expression and DNA replication through a peroxisome proliferator-activated receptor coactivator 1-alpha (PGC-1α)-dependent pathway in HCC cells [18]. In addition, GP could significantly recover DMN-induced fibrosis in mice and reverse 67% genes to a nearly normal state, as well as mitigate chemical-induced hepatic damage and fibrosis in vivo and suppress HSC and Kupffer cell activation in vitro [19,20]. However, its effectiveness and mechanism in protection against liver injury had not been elucidated. 

Here, we demonstrate a more efficient extraction method for the hepatoprotective effects of GP in cell and animal models of hepatotoxicity. We found that the 30% DMSO GP extract exhibited more cytotoxic activity in the activated hepatic stellate cell lines HSC-T6 and LX-2 than that of the methanolic extract. Next, a partially purified 30% DMSO GP extract, HH-F3, significantly inhibited the TGF-β-induced proliferation and migration of activated HSCs. Moreover, HH-F3 alleviated liver injury in a DEN-induced mouse model.

## 2. Results

### 2.1. The Effects of Different GP Preparations on HSC-T6 Cells

To test the potential biological effects of GP, various GP extracts prepared using different solvents, including water, butanol, acetone, 70% methanol, 95% ethanol, 70% ethanol, 50% ethanol, 100% DMSO, and 30% DMSO were used to treat HSC-T6 cells. The growth inhibitory effects of the GP extracts caused by different solvent preparations were evaluated. Data from the MTT assay indicated that the 30% DMSO and 70% methanol extracts significantly reduced the viability of HSC-T6 cells (Figure 1A); the 50% inhibition of cell viability (IC_50_) values of 30% DMSO extract 48 h posttreatment in HSC-T6 cells was 366 μg/mL (Figure 1B). Moreover, the 30% DMSO GP extract could significantly inhibit the protein expression of Aurkb in activated hepatic stellate cells (HSC-T6) (Figure 1C), suggesting that the active component(s) of GP exists in this extract. Previously, using a Sephadex LH-20 column, four fractions (HH-F1, HH-F2, HH-F3, and HH-F4) were obtained from the 30% DMSO GP extract [17].

### 2.2. HH-F3 Reduces the Viability of Hepatic Stellate Cells and Has No Cytotoxic Effect on Hepatocytes

To investigate the effects of HH-F3 on cell viability, HSC-T6 and LX-2 cells were treated with HH-F3 at concentrations of 0 μg/mL, 5 μg/mL, 25 μg/mL, 50 μg/mL, 75 μg/mL, and 100 μg/mL for 24, 48, and 72 h. The IC_50_ values of HH-F3-treated HSC-T6 and LX-2 cells at 72 h were determined to be 20.8 μg/mL and 22.5 μg/mL, respectively (Figure 2A,B). Moreover, HH-F3 could induce apoptosis in HSC-T6 and LX-2 cells via decreased mitochondrial membrane potential (Appendix A and Figure 2C). To examine the cytotoxic effect of the 30% DMSO GP extract and HH-F3 on primary cultured mouse hepatocytes, the MTT assay was used, and the results showed that both the 30% DMSO GP extract and HH-F3 treatments had no cytotoxic effect on primary cultured mouse hepatocytes (Appendix A).

### 2.3. GP/HH-F3 Inhibits TGF-β1-Induced HSC-T6 Cell Proliferation

TGF-β1 is a key mediator and profibrogenic cytokine that regulates the expression of collagen in the pathogenesis of liver fibrosis, whereas α-SMA reflects HSC activation [1,21,22]. A previous study showed that TGF-β1 influences microtubule stability, stabilizes aurora kinase B (AURKB), and induces AURKB expression and activation [23]. To examine whether the 30% DMSO GP extract and HH-F3 could inhibit the TGF-β1-triggered fibrotic effects on cell viability and ECM production in HSC-T6 cells, HSC-T6 cells were treated with TGF-β1 at doses of 0 ng/mL, 0.5 ng/mL, 1 ng/mL, 2 ng/mL, and 3 ng/mL for 0, 3, 6, 12, 24, and 48 h. TGF-β1-induced collagen I and α-SMA expression were determined at the protein level by Western blot analysis. As shown in Appendix A, treatment with TGF-β1 stimulated collagen I and α-SMA protein expression in a time-dependent and dose-dependent manner.

To further investigate the influence of the 30% DMSO GP extract and HH-F3 on the TGF-β1-induced proliferation of HSC-T6 and LX-2 cells, cells were first incubated with 1 ng/mL of TGF-β1 for 1 h; then, the 30% DMSO GP extract or HH-F3 was added at different concentrations for another 24, 48, and 72 h. The SRB assay was performed to assess cell proliferation. The results demonstrated that TGF-β1 treatment alone significantly stimulated the proliferation of HSC-T6 cells. However, treatment with the 30% DMSO GP extract and treatment with HH-F3 with or without TGF-β1 pretreatment dramatically inhibited the proliferation of HSC-T6 cells (Figure 3A,B and Appendix A).

### 2.4. GP/HH-F3 Inhibit the TGF-β1-Induced HSC Activation of Collagen Matrix Expression and the TGF-β Signaling Pathway

To elucidate the effects of the 30% DMSO GP extract and HH-F3 on the TGF-β1-induced expression of α-SMA and ECM proteins in HSC-T6 cells, Western blot analysis was performed. The addition of the 30% DMSO GP extract and HH-F3 significantly inhibited TGF-β1-induced collagen I, collagen III, elastin, and α-SMA protein expression in a dose-dependent manner (Figure 3C and Appendix A).

The SMAD-dependent pathway and SMAD-independent pathway (MAPK pathway) are key pathways leading to liver fibrosis and are activated by TGF-β1 [24,25]. Therefore, the protein levels of p-SMAD2, p-SMAD3, and p-P38 were determined in TGF-β1-stimulated HSC-T6 cells. The phosphorylation levels of SMAD2, SMAD3, and P38 increased in TGF-β1-treated HSC-T6 cells, whereas the expression of total SMAD2, SMAD3, and P38 proteins remained unchanged (Figure 3D and Appendix A). Treatment with the 30% DMSO GP extract and HH-F3 significantly decreased the levels of phosphorylated SMAD2, SMAD3, mitogen-activated protein kinase kinase (MEK), and P38 (Figure 3D and Appendix A).

### 2.5. HH-F3 Inhibits TGF-β1-Induced Cell Migration/Invasion

A previous study showed that the phosphorylation of MAPKs is involved in the activation and migration of cells [26]. Accumulating evidence shows that P38 activation leads to cell migration [27]. Thus, the effect of HH-F3 on HSC-T6 cell motility was observed by a wound-healing assay. The results indicated that HH-F3 markedly retarded the migration of HSC-T6 cells (Figure 4A). Next, the inhibition of cell migration/invasion by HH-F3 was further confirmed by a Transwell invasion assay. Similarly, HH-F3 effectively inhibited cell migration/invasion in a dose-dependent manner, and an ~15%, 37%, and 48% reduction was observed in the HSC-T6 cells treated with 5 μg/mL, 10 μg/mL, and 15 μg/mL of HH-F3, respectively (Figure 4B).

### 2.6. HH-F3 Effectively Alleviates DEN-Mediated Liver Injury and Fibrosis In Vivo

To further elucidate the antifibrotic effects of HH-F3 in vivo, the DEN-induced chronic liver injury in rats was determined. DEN is a well-known chemical carcinogen that stimulates the hepatic inflammation-fibrosis-cancer axis [28]. In this animal disease model, the degree of cirrhosis in rats was evaluated by serum and histopathological factors. As shown in Figure 5, DEN led to significantly increased levels of the biochemical markers ALT, AST, and γ-GT compared to the normal control group, indicating hepatocyte damage, while reductions in ALT, AST, and γ-GT were observed in the HH-F3-treated and sorafenib-treated (positive control) [29,30] groups compared to DEN treatment. Data from histological immunostaining showed that a large number of α-SMA-positive cells (activated HSCs) were observed in the fibrous septa, portal tracts, and sinusoids of the livers from DEN-treated rats (Appendix A). However, in all the drug-treated groups, including sorafenib and HH-F3, there was a significantly reduced number of activated HSCs (Appendix A). Moreover, a total protein sample was prepared from the liver tissues of the tested rats, and immunoblotting was performed. The effect of HH-F3 on fibrosis and the TGF-β1-regulated pathway were examined in an in vivo model. Figure 6A shows that DEN treatment increased the levels of collagen I, collagen III, and α-SMA in the rat liver compared to the normal rat liver. Treatment with HH-F3 significantly reduced the expression levels of these proteins, suggesting that HH-F3 could effectively inhibit fibrogenesis (Figure 6A). We further analyzed the TGF-β1-activated SMAD-dependent pathway and MAPK pathway through Western blot analysis. The hepatic expression of phosphorylated SMAD2 and p38 was significantly higher in DEN-treated livers compared to normal livers; however, the administration of HH-F3 significantly decreased the expression of these proteins (Figure 6B). These results suggest that HH-F3 could alleviate DEN-induced hepatic fibrosis through the interruption of the TGF-β signaling that triggered the activation and/or proliferation of HSCs.

## 3. Discussion

Chronic hepatic injury commonly leads to liver fibrosis, resulting in the development of liver cirrhosis and the consequent failure of hepatic function. Major cascades of the injury process have been reported, including interactions between damaged hepatocytes and Kupffer cell activation, HSC proliferation, and activation and the excessive production of ECM components, mainly type I collagen, during hepatic fibrosis [31]. In response to chronic liver injury, quiescent HSCs are activated to transdifferentiate into myofibroblasts, which are capable of the expression and secretion of almost all the connective tissue elements (collagens, elastin, structural glycoproteins, proteoglycans, and hyaluronan) representing the matrix of the fibrotic liver, the release of vitamin A, and the acquisition of contractility [32,33]. The activation of HSCs is considered a key factor in liver fibrosis progression; therefore, the prevention of HSC activation is a promising strategy for the treatment of hepatic fibrosis. Previously, several herbal medicines, such as phyllanthus, silymarin, glycyrrhizin, and sho-saiko-to, were reported to have antifibrotic properties [34,35,36,37]. Sal B, the active component of *Salvia miltiorrhiza*, protects hepatocytes against CCl_4_-induced cell damage, lipid peroxidation, and ROS formation. Sal B also attenuates TGF-β1-stimulated α-SMA expression in rat HSCs [38]. GP is a commonly used herbal medicine in Taiwan, and it has been reported that GP extracts have antifibrotic and anti-inflammatory effects both in vivo and in vitro [15,19,20]. This study showed that GP/HH-F3 exhibited antifibrotic effects on in vivo DEN-injured rat livers by reducing (a) plasma levels of ALT and AST, (b) hepatic α-SMA protein accumulation, (c) hepatic fibrosis score and collagen deposition, and (d) the TGF-β pathway. An in vitro study demonstrated that GP/HH-F3 suppressed HSC activation via the inhibition of TGF-β1-induced proliferation, migration, and invasion. Overall, HH-F3 from *Graptopetalum paraguayense* (GP) significantly suppressed hepatic fibrosis in DEN-injured rats and inhibited HSC activation, which is the key event associated with hepatic fibrosis.

It is now widely accepted that the pro-inflammatory cytokine TGF-β1 is a major cytokine involved in the regulation of the production, degradation, and accumulation of the ECM during liver fibrosis progression [39,40]. Increasing evidence shows that TGF-β1 is a key mediator in the pathogenesis of liver fibrosis in both experimental and human liver diseases: TGF-β1 is highly upregulated in the form of either the transcript or protein in the fibrotic tissues of animal models or human samples [41,42]. TGF-β1 mediates progressive liver fibrosis by stimulating ECM production while inhibiting its degradation [43,44]. Therefore, TGF-β1 is regarded as a potent multifunctional cytokine that is believed to be a crucial mediator in chronic liver injury. TGF-β/SMAD signaling has been shown to be an important downstream intracellular signaling pathway. TGF-β also activates SMAD-independent signaling pathways in part by increasing ROS production, leading to the activation of other signaling cascades such as MAPK and the upregulation of fibrogenic genes [45]. Blocking the TGF-β signaling pathway has been proposed as a promising therapeutic intervention in chronic liver injury [46]. α-SMA and collagen I, which are ECM proteins produced by typical myofibroblastoid cells as a wound-healing response, are stimulated by SMAD-dependent TGF-β signaling [47]. Furthermore, TGF-β and collagen I can upregulate HSC migration [48]. This study showed that GP/HH-F3 could suppress the TGF-β1-induced SMAD-dependent pathway and the SMAD-independent pathway (P38 and MEK). Furthermore, GP/HH-F3 not only attenuated TGF-β1-induced HSC migration, but also inhibited α-SMA and collagen I expression.

In the DEN model, the sequence of events leading to steatohepatitis, fibrosis, cirrhosis, and tumors is completely skipped; therefore, this liver disease model appears to be a suitable animal model for the analysis of some aspects and processes that promote the pathogenesis of liver fibrosis, cirrhosis, and hepatocellular carcinoma [49,50]. Previous studies have shown that enhanced TGF-β expression is observed in the liver as early as in the fourth week of DEN administration [28]. TGF-β normally acts as a mitogenic factor in liver regeneration, and this probably occurs in DEN-treated livers. Overactivation of the TGF-β/epidermal growth factor receptor (EGFR) signaling pathway also leads to hepatocarcinogenesis [51]. Moreover, DEN-induced liver fibrosis in rats could increase the phosphorylation of JNK1/2, p38, and ERK1/2 [52]. Our previous study indicates that GP/HH-F3 treatment can overcome DEN-induced liver cancer [17]. The current study showed that HH-F3 effectively decreased hepatic collagen levels and diminished DEN-induced liver fibrosis in rat models. In the animal models, we also found that HH-F3 could meliorate liver fibrosis via the inhibition of TGF-β-mediated ECM expression as well as the TGF-β-regulated SMAD-dependent pathway and SMAD-independent pathway. The administration of GP/HH-F3 significantly inhibited the DEN-induced serum markers AST, ALT, and γ-GT; it also decreased α-SMA and collagen expression, alleviated liver fibrosis, and largely improved liver functional injury in rats.

In conclusion, our observations demonstrated that GP/HH-F3 ameliorated DEN-induced liver injury and inhibited fibrogenesis in rats through the regulation of HSC activation by the inhibition of the TGF-β pathway. These findings indicate that GP/HH-F3 might be considered a potential therapeutic agent in the treatment of liver fibrosis.

## 4. Methods

### 4.1. GP Extraction and HH-F3 Purification

The leaves of GP were ground and lyophilized into a powder at –20 °C and stored in a moisture buster at 25 °C before extraction. The details of GP extraction were defined in our previous study [17]. The 30% DMSO supernatant was either stored at –20 °C as a 150 mg/mL stock solution (referred to as 30% DMSO GP extracts) or fractionated into four fractions (F1–F4) by a Sephadex LH-20 column. Using the analysis of AURKA, AURKB, and FLJ10540 protein levels via Western blot, active components were obtained in Fraction (3) and referred to as HH-F3. HH-F3 was further analyzed by matrix-assisted laser desorption ionization-time of flight (MALDI-TOF) mass analysis, confirming these major peaks in the MALDI-TOF at approximately 4.5 kD, 7.1 kD, and 8.9 kD. Phloroglucinol degradation was used to analyze the components that constitute the active fraction of HH-F3. Briefly, an HH-F3 sample was hydrolyzed in the presence of acid and phloroglucinol, and the degradation products were then analyzed by LC/MS. The active fraction contains polymeric tannins that comprise one or more phenolic compounds selected from the group consisting of (–)-epicatechin, (–)-epigallocatechin, (–)-epigallocatechin gallate, (–)-epicatechin-3-O-gallate, and (–)-epigallocatechin gallate. The main components include polymeric tannins that are composed of compounds having (–)-epigallocatechin gallate as the terminal units and compounds having (–)-epigallocatechin gallate as the main units, as shown in Figure 7. The samples were stored at −20 °C.

### 4.2. Cell Culture

Rat HSC-T6 and human LX-2 cell lines were obtained from Dr. Yun-Lian Lin at The National Research Institute of Chinese Medicine, Taiwan. HSC-T6 cells, immortalized early active stage rat hepatic stellate cells (HSCs), were maintained in Waymouth MB 752/1 medium (Sigma-Aldrich, St. Louis, MO, USA) supplemented with 10% fetal calf serum and antibiotics (100 IU/mL each of penicillin and streptomycin) in a humidified atmosphere containing 5% CO_2_ and 95% air at 37 °C. LX-2, immortalized early active stage human HSCs were maintained in Dulbecco’s Modified Eagle medium (DMEM) supplemented with 2% fetal calf serum and antibiotics (100 IU/mL each of penicillin and streptomycin) in a humidified atmosphere containing 5% CO_2_ and 95% air at 37 °C. The cultures were passaged by trypsinization every 3–4 days.

### 4.3. MTT Assay

Cells were seeded in 24-well plates (4000–5000 cells/well), incubated overnight, and then treated with the 30% DMSO GP extract or HH-F3 for 0, 24, 48, or 72 h. The viability of the cells in the medium containing FBS was stable for the experimental period. After treatment, the cells were gently washed three times with 1× PBS and then incubated with 0.5 μg/mL 3-(4,5-dimethylthiazol-2-yl)-2,5-diphenyl tetrazolium bromide (MTT) (Sigma-Aldrich, St. Louis, MO, USA) for 2 h. The medium was removed, and the deep-blue crystals were dissolved in 100% DMSO at room temperature for 10 min. optical density (OD) values were measured at 570 nm with an ELISA reader.

### 4.4. Cell Counting

The cells were seeded in 12-well plates (10,000-30,000 cells/well) overnight and then treated with HH-F3 for 0, 24, 48, or 72 h. The cells were trypsinized and then counted after mixing with 0.4% trypan blue.

### 4.5. Sulforhodamine B (SRB) Assay

Cells were seeded into plates, cultured overnight, and then treated with GP or HH-F3 for 24 h. The medium was removed immediately after drug treatment and replaced with 10% precooled trichloroacetic acid (TCA). The cells were fixed for 1 h at 4 °C; then, the plate was washed with distilled water and dried. One hundred microliters of a 4 mg/mL solution of SRB (Sigma-Aldrich, St. Louis, MO, USA) in 1% acetic acid was added to each well, and the plate was incubated for 1 h. The plate was then washed five times with a 1% acetic acid solution and dried. The SRB in the cells was subsequently dissolved in 150 μL of 10 mM Tris-HCl and measured at 540 nm using an ELISA reader. 

### 4.6. Western Blot

All the samples were denatured by heating at 95 °C for 10 min and resolved by 8%, 10%, or 12% SDS-polyacrylamide gel electrophoresis (SDS-PAGE) at 80 V and 100 V for the stacking and running gels, respectively. After SDS-PAGE, the proteins were transferred to polyvinylidene difluoride (PVDF) membranes using the Bio-Rad transfer system. After the proteins were transferred, the membranes were stained with Ponceau S to confirm the efficiency and uniformity of the protein transfer. The membranes were blocked with 5% nonfat skim milk at room temperature for 30 min, incubated with the primary antibody at 4 °C overnight, and then washed with 1× Tris-buffered saline Tween-20 (TBST) three times (10 min each). The membranes were incubated with the secondary antibody for 2 h. Then, they were washed with 1× TBST three times (10 min each). The signals from the secondary antibodies were visualized by adding horseradish peroxidase (HRP) substrate peroxide solution/luminol reagents (ImmobilonTM Western Chemiluminescent Substrate, Millipore; mixed at a 1:1 ratio) and were detected by a Fujifilm LAS 4000 luminescent image analysis system.

### 4.7. Transwell Cell Migration/Invasion Assay

The migration/invasion assay was based on a previously reported method [53]. The migration/invasion of HH-F3-treated cells was evaluated using a 24-well Transwell (8-mm pore size polycarbonate membrane, Millipore) chamber. Cells were trypsinized and suspended in Waymouth MB 752/1 medium containing 0.05% FBS. A total of 10^5^ cells in 0.2 mL of Waymouth MB 752/1 medium, containing 0.05% FBS without or with various concentrations of HH-F3, were plated in the insert chambers. Waymouth MB 752/1 medium containing TGF-β1 (1 ng/mL) and 0.05% serum was added as a chemoattractant in the outside chambers. After 10 h of incubation at 37 °C, the migratory cells were fixed in 100% ice-cold methanol and stained with Giemsa stain (1:5 in ddH_2_O, Sigma-Aldrich), and the cells on the upper surface of the membrane were mechanically removed with a cotton swab. The membranes were photographed by microscopy with a camera. Cell migration values were determined by counting 3 to 5 fields per filter membrane, and then normalized using negative control cells to give a relative ratio.

### 4.8. Wound-Healing Assay

To investigate the migration of HSCs, the wound-healing assay was performed according to a previously reported method [54,55]. Briefly, for the in vitro cell migration assay, a culture insert (ibidi GmbH, Martinsried, Germany) was used to create a wound in the cell culture. HSC-T6 cells were seeded onto cell-culture inserts in 24-well culture plates at a density of 1×10^4^ cells/well and allowed to grow to confluency. The HSC-T6 cells were incubated at 37 °C for 24 h and then exposed to HH-F3 (5, 10, or 15 μg/mL) and TGF-β1 (1 ng/mL) in culture media containing 2% FBS for cell migration analysis. Cell debris was removed by washing the coverslips once with sterile PBS, and cells were fixed by incubation with 80% ethanol for 10 min. The cells were stained with crystal violet, and migrating cells were counted.

### 4.9. Animal Model

The 80 Wistar rats were randomly divided into five groups: a normal group, a diethylnitrosamine (DEN) group, a sorafenib group, a low-dose HH-F3 group, and a high-dose HH-F3 group. Sorafenib was used as the positive control [29,30]. In all the groups except the normal group, the rats drank an aqueous solution of 100 ppm (*v/v*) DEN (Sigma-Aldrich, St. Louis, MO, USA) daily as the sole source of drinking water for 63 days, and starting on day 28, they were fed tap water for another 42 days. DEN solution was prepared each week and consisted of an individualized dose according to the weight gain/loss of the animal in response to the previous dose. During the experimental period, the animals were weighed weekly to calculate weight gain, and the amount of water consumed was also measured every week. In the sorafenib group, the rats received 30 mg/kg/rat of sorafenib powder. In the low-dose group, the rats received 50 mg/kg/rat of lyophilized HH-F3 powder, and in the high-dose group, the rats received 150 mg/kg/rat of lyophilized HH-F3 powder daily. All the rats were sacrificed on the 70^th^ day. 

### 4.10. Ethics Statement

All the animal procedures followed the Guide for the Care and Use of Laboratory Animals published by the United States (US) National Institutes of Health (NIH publication No. 85–12, revised 1996) and animal care guidelines of National Taiwan University, and were approved by the Animal Research Committee at the National Taiwan University under IACUC protocol no: 20090352.

### 4.11. Hepatic Enzyme Measurement

Serum alanine aminotransferase (ALT), aspartate aminotransferase (AST), and γ-Glutamyltransferase (γ-GT) levels were determined using enzyme linked immunosorbent assay methods according to the kit protocols (Cayman Chemical Co., Ann Arbor, MI, USA).

### 4.12. Histological Analysis

The liver samples were fixed with formalin, embedded with paraffin, and then sectioned into 5-μm sections. The sections were deparaffinized, rehydrated, and then treated with 0.03% hydrogen peroxide for 10 min to quench endogenous peroxidase activity. Following two washes with PBS, the sections were incubated for 1 h at room temperature with a mouse anti-alpha-smooth muscle actin (α-SMA) monoclonal antibody (1:50 dilution, DakoCytomation, Glostrup, Denmark). For α-SMA staining, the sections were washed and further incubated with a secondary antibody, rabbit anti-mouse IgG (1:200 dilution), at room temperature for 1 h. Then, the sections were developed similarly. After staining, the sections were counterstained with hematoxylin for microscopic examination. The percentage of the α-SMA-positive area (mm^2^/cm^2^ of liver section) was measured using the Digital Camera System HC-2500 (Fuji Photo Film, Tokyo, Japan), Adobe Photoshop version 5.0J, and Image-Pro Plus version 3.0.1J.

### 4.13. Statistical Analysis

All the results are expressed as the mean ± standard error of the mean. Levels of significance were evaluated by two-tailed paired Student’s *t*-test or one-way analysis of variance (ANOVA). *p* < 0.05 was considered statistically significant.

## Figures and Tables

**Figure 1 ijms-20-02592-f001:**
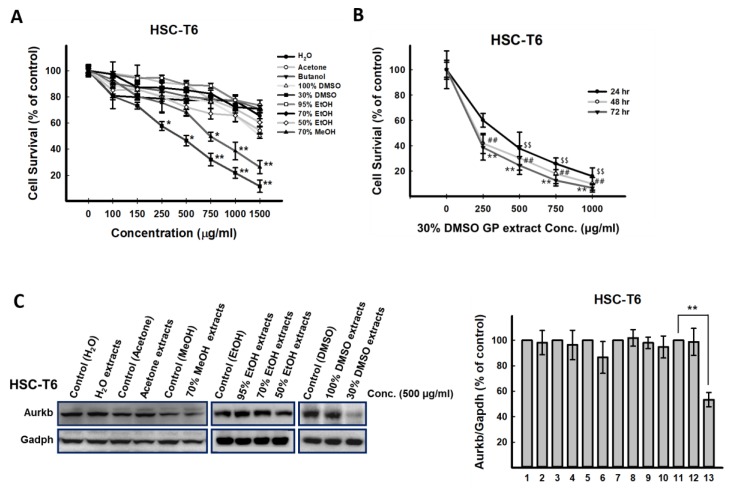
Treatment with the 30% DMSO *Graptopetalum paraguayense* (GP) extract, but not other extracts prepared from different solvents, suppressed Aurkb protein expression in HSC-T6 cells. (**A**) HSC-T6 cells were treated with GP extracts prepared from water, butanol, acetone, 70% methanol, 100% ethanol, 70% ethanol, 50% ethanol, 100% DMSO, and 30% DMSO at concentrations of 100 μg/mL, 150 μg/mL, 250 μg/mL, 500 μg/mL, 750 μg/mL, 1000 μg/mL, and 1500 μg/mL for 72 h, after which the cells were subjected to 3-(4,5-dimethylthiazol-2-yl)-2,5-diphenyl tetrazolium bromide (MTT) assays. * *p* < 0.05, ** *p* < 0.01, compared with the control group. (**B**) HSC-T6 cells were treated with the 30% DMSO GP extract at concentrations of 0 μg/mL, 250 μg/mL, 500 μg/mL, 750 μg/mL, and 1000 μg/mL for 24, 48, and 72 h, followed by MTT assays. ^$^
*p* < 0.05, ^$$^
*p* < 0.01, compared with the control group (24 h); ^#^
*p* < 0.05, ^##^
*p* < 0.01, compared with the control group (48 h); * *p* < 0.05, ** *p* < 0.01, compared with the control group (72 h). (**C**) HSC-T6 cells were treated with GP extracts in H_2_O, acetone, 70% methanol, 100% ethanol, 70% ethanol, 50% ethanol, 100% DMSO, and 30% DMSO at 500 μg/mL for 24 h. The protein expression of Aurkb was detected by Western blot. 1: Control (H_2_O), 2: H_2_O extract, 3: Control (acetone), 4: acetone extract, 5: Control (methanol), 6: 70% methanol extract, 8: Control (ethanol), 9: 100% ethanol extract, 10: 70% ethanol extract, 9: 50% ethanol extract, 11: Control (DMSO), 12: 100% DMSO extract, 13: 30% DMSO extract. * *p* < 0.05, ** *p* < 0.01, compared with the control group.

**Figure 2 ijms-20-02592-f002:**
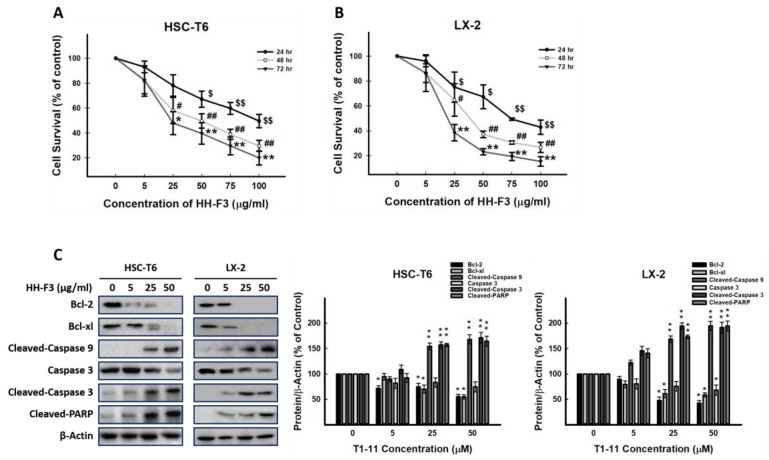
HH-F3 inhibits the growth of hepatic stellate cells via apoptosis. (**A**) HSC-T6 and (**B**) LX-2 cells were treated with 5 μg/mL, 25 μg/mL, 50 μg/mL, 75 μg/mL, and 100 μg/mL of HH-F3 for 24, 48, and 72 h and then subjected to MTT assays. ^$^
*p* < 0.05, ^$$^
*p* < 0.01, compared with the control group (24 h); ^#^
*p* < 0.05, ^##^
*p* < 0.01, compared with the control group (48 h); * *p* < 0.05, ** *p* < 0.01, compared with the control group (72 h). (**C**) HSC-T6 and LX-2 cell lysates treated with HH-F3 (24 h) were subjected to immunoblot analysis for anti-cleaved caspase 9, anti-cleaved caspase-3 and cleaved PARP.

**Figure 3 ijms-20-02592-f003:**
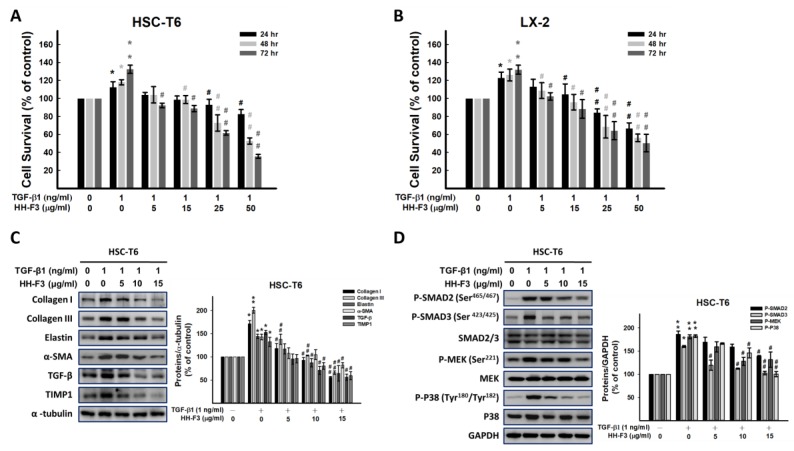
HH-F3 inhibits TGF-β1-induced cell proliferation via TGF-β pathway suppression. (**A**) HSC-T6 and (**B**) LX-2 cell viability was assayed in the presence of HH-F3 and transforming growth factor beta (TGF-β1, 1 ng/mL) by SRB assay. (**C**) HSC-T6 cells were preincubated with 1 ng/mL of TGF-β1 for 1 h and then cotreated with HH-F3 at different concentrations. The dose-dependent inhibitory effect on collagen type I, collagen type III, elastin, and α-SMA expression was determined by Western blot analysis. Data represent at least three independent experiments. (**D**) HSC-T6 cells were pretreated with 1 ng/mL of TGF-β for 1 h and then cotreated with various concentrations of HH-F3 for 24 h. HH-F3 blocked TGF-β1 (1 ng/mL)-induced SMAD2/3, MEK, and P38 phosphorylation. Data represent at least three independent experiments. * *p* < 0.05, ** *p* < 0.01 compared to the control group; ^#^
*P* < 0.05 compared to the TGF-β1-induced group.

**Figure 4 ijms-20-02592-f004:**
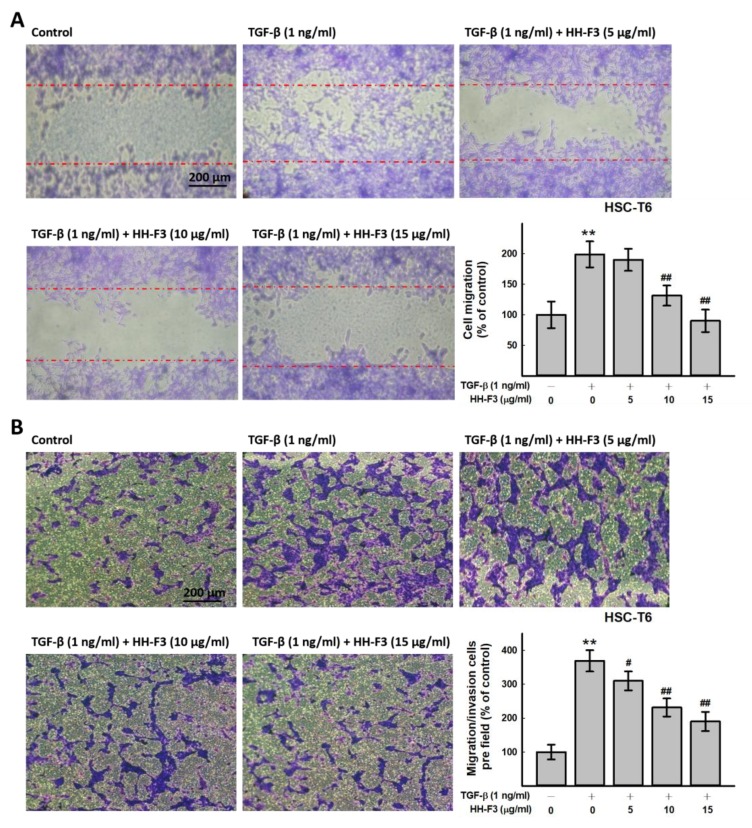
HH-F3 affects the migration ability of TGF-β-activated HSC-T6 cells. Cell migration was monitored by using wound-healing and transwell invasion assays. (**A**) For wound-healing assays, cells were seeded into culture inserts. HSC-T6 cells were pretreated with 1 ng/mL TGF-β1 and then treated with 5 μg/mL, 10 μg/mL, or 15 μg HH-F3 for 24 h. Red dotted lines are started wound edges. (**B**) For migration/invasion assays, cells were seeded into the top of a transwell insert. HSC-T6 cells were cotreated with 1 ng/mL of TGF-β1 and 5 μg/mL, 25 μg and 50 μg/mL HH-F3 for 24 h. After eight hours, the cells on the top were scraped, and the cells that had migrated to the bottom were fixed and stained with Giemsa. The relative-fold migration values were normalized against the negative control and are represented diagrammatically. ** *p* < 0.01 compared to the control group; ^#^
*p* < 0.05, ^##^
*p* < 0.01 compared to the TGF-β1-induced group.

**Figure 5 ijms-20-02592-f005:**
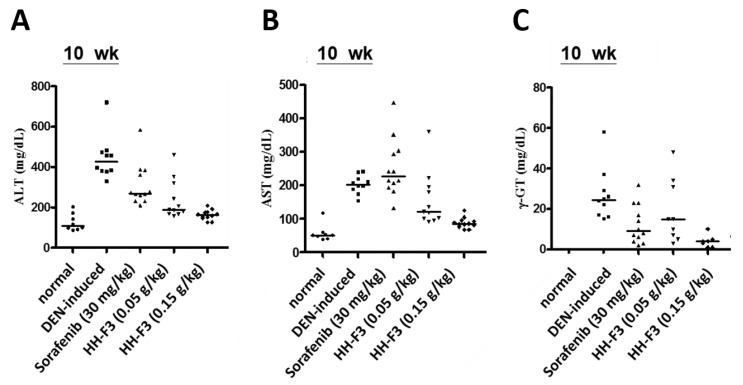
In vivo effects of HH-F3 on diethylnitrosamine (DEN)-induced liver fibrosis increases in serum biomarkers. The rats were divided into five groups: group 1, normal control (no treatment); group 2, model group (DEN operation); group 3, sorafenib (30 mg/kg) and DEN-treated group; group 4, HH-F3 (0.05 g/kg) and DEN-treated group; group 5, HH-F3 (0.15 g/kg) and DEN-treated group. (**A**) Serum alanine aminotransferase (ALT), (**B**) aspartate aminotransferase (AST), and (**C**) γ-glutamyltransferase (γ-GT) levels in mice. Levels of serum ALT, AST, and γ-GT were significantly decreased to nearly the normal level after antifibrosis treatment with high-dose HH-F3.

**Figure 6 ijms-20-02592-f006:**
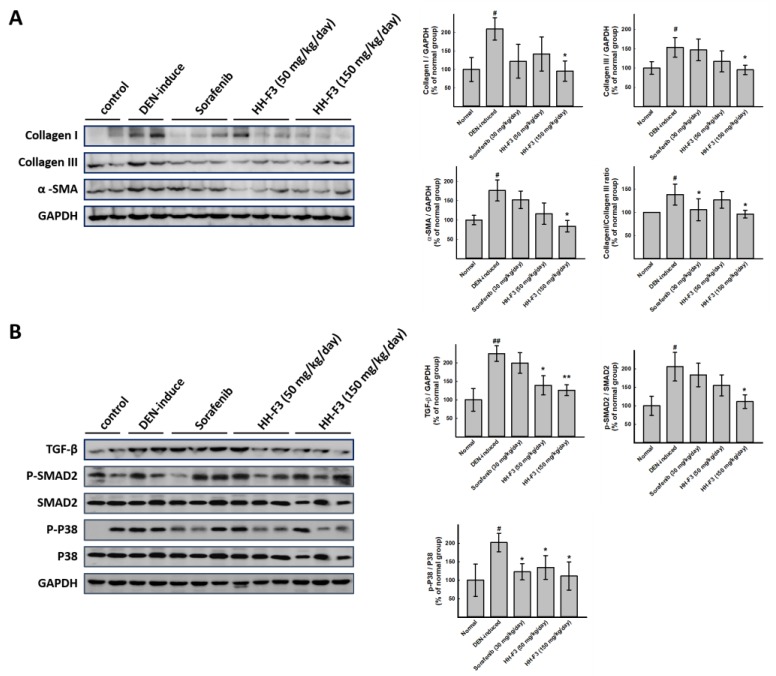
Effects of HH-F3 on fibrogenesis markers and the TGF-β pathway from the in vivo model. Rats were divided into five groups. One group served as the untreated control, one group was treated with DEN to induce liver fibrosis, and the other groups were cotreated with DEN and sorafenib or different doses of HH-F3 (50 mg/kg/day and 150 mg/kg/day) orally (oral administration, P.O.) every day. Rats were sacrificed and the liver tissues were homogenized followed by Western blot analysis and quantification. (**A**) HH-F3 reduced fibrogenesis markers in a DEN-induced liver fibrosis model. (**B**) HH-F3 inhibited the TGF-β pathway in a DEN-induced liver fibrosis model. * *p* < 0.05, ** *p* < 0.01 compared to the DEN group; ^#^
*p* < 0.05 compared to the normal group.

**Figure 7 ijms-20-02592-f007:**
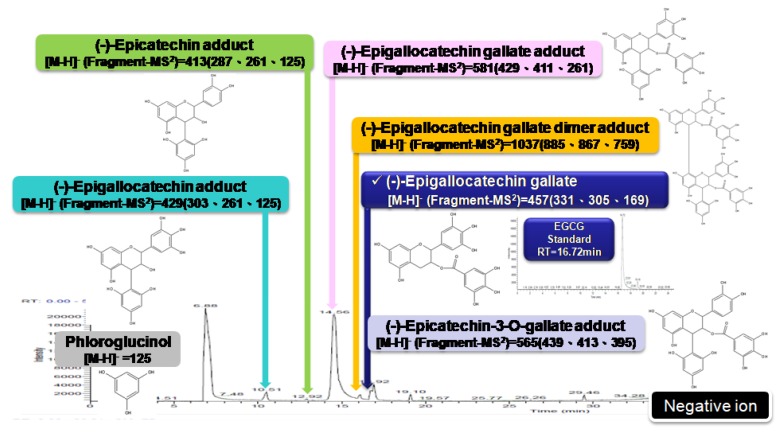
Phloroglucinol degradation for HH-F3 (LC/MS profile).

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
