# Peer review of "Graptopetalum paraguayense Inhibits Liver Fibrosis by Blocking TGF-β Signaling In Vivo and In Vitro"

_ijms, 2019, doi:10.3390/ijms20102592_

Round 1
Reviewer 1 Report
The article “Graptopetalum paraguayense Inhibits Liver Fibrosis by Blocking TGF-β Signaling In Vivo and In Vitro” is interesting and properly prepared.
The "Introduction" section introduces the topic of work well.
The "Materials and Methods" section is generally good, and I do not have any comments.
The section "Results" is interesting and I have only minor comments
1. The authors should add densitometry and statistical analysis in Figure 2C and 3C
2. The representative images of migration should be added in Figure 5 to improve its readability.
3. Why collagen III level was analyzed? Could the authors write some information supporting that chose?
The authors summarized the results and compared them with the research of other teams. The conclusions drawn from the data analysis are correct. All results have been discussed, so I do not have any comment for that part.
Author Response
Response to Reviewer comments for ijms-498901
Title: Graptopetalum paraguayense Inhibits Liver Fibrosis by Blocking TGF-β Signaling In Vivo and In Vitro
Author(s): Wei-Hsiang Hsu , Se-Chun Liao , Yau-Jan Chyan , Kai-Wen Huang , Shih-Lan Hsu , Yi-Chen Chen , Ma-Li Siu , Chia-Chuan Chang , Yuh-Shan Chung , Chi-Ying F. Huang
Dear Editor:
We would like to thank the reviewers for their thorough reading of our manuscript as well as their valuable comments. We have conscientiously re-examined the manuscript and made certain revisions. Enclosed herewith are the itemized changes. We have followed their comments closely and feel that their suggestions have further strengthened the manuscript. Below are our point-by-point responses.
Reviewer comments:
Reviewer #1 (Remarks to the Author):
Comment 1. The authors should add densitometry and statistical analysis in Figure 2C and 3C
Response: Thanks for your suggestion. We have added densitometry and statistical analysis of Figure 2C and 3C in our manuscript.
Comment 2. The representative images of migration should be added in Figure 5 to improve its readability.
Response: As suggested, we have added representative images of migration in Figure 5.
Comment 3. Why collagen III level was analyzed? Could the authors write some information supporting that chose?
Response: Thanks for this valuable comment. Collagen III is one of major compositions of extracellular matrix (ECM). A substantial change in ECM composition is the deposition of collagens, mainly fibril-forming types I, III, and IV which increase in fibrotic ECM. Many studies also showed that the levels of collagen types I, II, III, and IV increased significantly with the progression of fibrosis [1-3]. Besides, in the progression of fibrosis, the ratio of collagen types I/collagen type III will be increased [4, 5]. Thus, we detected and analyzed levels of both collagen type I and III, as well as calculated collagen I/collagen III ratio. We have made it clear in the manscript.
References
1. Schuppan, D.; Ruehl, M.; Somasundaram, R.; Hahn, E. G., Matrix as a modulator of hepatic fibrogenesis. Semin Liver Dis 2001, 21, (3), 351-72.
2. Hemmann, S.; Graf, J.; Roderfeld, M.; Roeb, E., Expression of MMPs and TIMPs in liver fibrosis - a systematic review with special emphasis on anti-fibrotic strategies. J Hepatol 2007, 46, (5), 955-75.
3. Baiocchini, A.; Montaldo, C.; Conigliaro, A.; Grimaldi, A.; Correani, V.; Mura, F.; Ciccosanti, F.; Rotiroti, N.; Brenna, A.; Montalbano, M.; D'Offizi, G.; Capobianchi, M. R.; Alessandro, R.; Piacentini, M.; Schinina, M. E.; Maras, B.; Del Nonno, F.; Tripodi, M.; Mancone, C., Extracellular Matrix Molecular Remodeling in Human Liver Fibrosis Evolution. PLoS One 2016, 11, (3), e0151736.
4. BENYON, R. C.; IREDALE, J. P., Is liver fibrosis reversible? Gut 2000, 46, (4), 443-446.
5. Rojkind, M.; Giambrone, M. A.; Biempica, L., Collagen types in normal and cirrhotic liver. Gastroenterology 1979, 76, (4), 710-9.
Reviewer 2 Report
The article of Hsuet al. entitled "Graptopetalum paraguayense inhibits liver fibrosis by blocking tif-beta signaling in vivo and in vitro" focused on the excessive accumulation of extracellular matrix proteins in chronic liver diseases. Based on previous studies of the same team, they test the potential mechanism of GP extracts and then its fraction, called HH-F3. They found that GP/HH-F3 can decrease collagen and some metabolites activities in rat liver models suggesting a potential therapeutic role for this agent in liver fibrosis.
The manuscript is clear and well written. The abstract is clear. Results are concordant with the methodology used by the authors. Discussion of the results is ok and the conclusion is concordant with the results.
Some comments may improve the accuracy of the manuscript:
- please more detail the definition of GP in introduction, and thus the different axes of action in the diseases mentioned (i.e. hypertension, cancer etc.) or mentioned that this is the same way as in fibrosis.
- Please more clearly wrote informations in figures (it is too small for the readers).
- please be careful to define the abbreviation from the beginning of their appearance;
- an article of IJMS could be interesting to add in your article: Vallée et al. "Thermodynamic aspects and reprogramming cellular energy metabolism during the fibrosis process". IJMS 2017.
Author Response
Response to Reviewer comments for ijms-498901
Title: Graptopetalum paraguayense Inhibits Liver Fibrosis by Blocking TGF-β Signaling In Vivo and In Vitro
Author(s): Wei-Hsiang Hsu , Se-Chun Liao , Yau-Jan Chyan , Kai-Wen Huang , Shih-Lan Hsu , Yi-Chen Chen , Ma-Li Siu , Chia-Chuan Chang , Yuh-Shan Chung , Chi-Ying F. Huang
Dear Editor:
We would like to thank the reviewers for their thorough reading of our manuscript as well as their valuable comments. We have conscientiously re-examined the manuscript and made certain revisions. Enclosed herewith are the itemized changes. We have followed their comments closely and feel that their suggestions have further strengthened the manuscript. Below are our point-by-point responses.
Reviewer comments:
Reviewer #2 (Remarks to the Author):
Comment 1. Please more detail the definition of GP in introduction, and thus the different axes of action in the diseases mentioned (i.e. hypertension, cancer etc.) or mentioned that this is the same way as in fibrosis.
Response: We are grateful for your suggestion. We rewrite and amend our paragraph about GP in the introduction of the manuscript.
Comment 2. Please more clearly wrote informations in figures (it is too small for the readers)
Response: As suggested, we have prepared the new figure to improve the resolution.
Comment 3. Please be careful to define the abbreviation from the beginning of their appearance
Response: Thanks for the comment. We have checked and corrected it.
Comment 4. An article of IJMS could be interesting to add in your article: Vallée et al. "Thermodynamic aspects and reprogramming cellular energy metabolism during the fibrosis process". IJMS 2017.
Response: Thank you for this information. We have added it as one of our references (reference 43) in manuscript.